# Epidemiology and pathological progression of erythematous lip lesions in captive sun bears (*Helarctos malayanus*)

Kirsty Officer[1,2]*, Mathieu Pruvot[3], Paul Horwood[4¤a], Daniela Denk[5], Kris Warren[2], Vibol Hul[4], Nhim Thy[6], Nev Broadis[1¤b], Philippe Dussart[4¤c], Bethany Jackson[2]

1 Free the Bears, Phnom Penh, Cambodia, 2 School of Veterinary Medicine, Murdoch University, Perth, Western Australia, Australia, 3 Health Program, Wildlife Conservation Society, Bronx, New York, United States of America, 4 Virology Unit, Institut Pasteur du Cambodge, Institut Pasteur International Network, Phnom Penh, Cambodia, 5 International Zoo Veterinary Group, Keighley, United Kingdom, 6 Forestry Administration, Ministry of Agriculture, Forestry and Fisheries, Phnom Penh, Cambodia

¤a Current address: College of Public Health, Medical and Veterinary Sciences, James Cook University, Townsville, Queensland, Australia
¤b Current address: Wildlife Conservation Society, Phnom Penh, Cambodia
¤c Current address: Institut Pasteur de Madagascar, Institut Pasteur International Network, Antananarivo, Madagascar
* kirsty.officer@murdoch.edu.au

**Data Availability Statement:** All relevant data are within the manuscript and its Supporting information files.

## Abstract

This study investigates the occurrence of erythematous lip lesions in a captive sun bear population in Cambodia, including the progression of cheilitis to squamous cell carcinoma, and the presence of Ursid gammaherpesvirus 1. Visual assessment conducted in 2015 and 2016 recorded the prevalence and severity of lesions. Opportunistic sampling for disease testing was conducted on a subset of 39 sun bears, with histopathological examination of lip and tongue biopsies and PCR testing of oral swabs and tissue biopsies collected during health examinations. Lip lesions were similarly prevalent in 2015 (66.0%) and 2016 (68.3%). Degradation of lip lesion severity was seen between 2015 and 2016, and the odds of having lip lesions, having more severe lip lesions, and having lip lesion degradation over time, all increased with age. Cheilitis was found in all lip lesion biopsies, with histological confirmation of squamous cell carcinoma in 64.5% of cases. Single biopsies frequently showed progression from dysplasia to neoplasia. Eighteen of 31 sun bears (58.1%) had at least one sample positive for Ursid gammaherpesvirus 1. The virus was detected in sun bears with and without lip lesions, however due to case selection being strongly biased towards those showing lip lesions it was not possible to test for association between Ursid gammaherpesvirus 1 and lip squamous cell carcinoma. Given gammaherpesviruses can play a role in cancer development under certain conditions in other species, we believe further investigation into Ursid gammaherpesvirus 1 as one of a number of possible co-factors in the progression of lip lesions to squamous cell carcinoma is warranted. This study highlights the progressively neoplastic nature of this lip lesion syndrome in sun bears which has consequences for captive and re-release management. Similarly, the detection of Ursid gammaherpesvirus 1 should be considered in pre-release risk analyses, at least until data is available on the prevalence of the virus in wild sun bears.

**Funding:** Laboratory testing at the Institut Pasteur du Cambodge was supported by funding from the European Union under the INNOVATE programme, through the LACANET project (DCIASIE/2013/315-151). The funders had no role in study design, data collection and analysis, decision to publish, or preparation of the manuscript.

**Competing interests:** The authors have declared that no competing interests exist.

## Introduction

The smallest of the extant bear species, the sun bear (*Helarctos malayanus*), primarily inhabits lowland forests in mainland Southeast Asia, Sumatra, and Borneo, and is threatened by habitat loss and hunting throughout its range [1]. Although infectious and non-infectious disease have been recognised as threats to wildlife elsewhere [2–5] there are no reports of disease in free-ranging sun bears, which may reflect the limited surveillance of this species in the wild. Reports of disease in captive sun bears can be found, including neoplasia [6–10], a small number of infectious disease reports [10–13], and, the focus of this study, progressive oral lesions [10]. Thus captive sun bears are an important source of health and disease data that ultimately may be applied to their wild counterparts, whether through risk analyses for future releases, or the provision of baseline data to inform study design for disease surveillance in wild sun bears.

Inflammation of the lips (cheilitis) has a range of reported causes in humans and other animals, including infectious, allergic, immune-mediated, climatic and contact irritant aetiologies, and can also be a manifestation of generalised skin or intra-oral conditions [14–16]. A substantial subset of human cheilitis cases are due to sun exposure and are considered pre-malignant lesions [17]. Of the infectious aetiologies, herpesviruses are associated with oral lesions in a number of species, including humans [18], macaques [19], reptiles including chelonians [20–23], northern sea otters (*Enhydra lutris kenyoni*) [24], elephants [25], and a captive fisher (*Martes pennant*) [26]. Certain herpes viruses (e.g. herpes simplex virus and Epstein-Barr virus) have been implicated in human oral squamous cell carcinoma (SCC) development [27], albeit their exact role in carcinogenesis may depend on additional co-factors. A key feature of the gammaherpesvirus sub-family is the capacity to induce lymphoproliferative diseases and other malignancies under certain conditions such as host immune compromise [28–32] through a variety of host cell manipulation mechanisms [33].

A novel gammaherpesvirus (Ursid gammaherpesvirus 1; UrHV-1) detected in four captive sun bears was first described by Lam et al. [10]. This newly identified herpesvirus was detected in oral SCC tissue samples, albeit the relationship between the virus and the neoplasia was unclear. More recently, closely related gammaherpesviruses were detected in tissues from free-ranging American black bears (*Ursus americanus*) with and without neurological signs [34]. Sequences of glycoprotein B and DNA polymerase genes from an ursid gammaherpesvirus strain (putatively designated as Ursid gammaherpesvirus 2) detected from a free-ranging Asiatic black bear (*Ursus thibetanus*) in Russia were added to the GenBank database in 2018 (MK089801).

Between 2011 and 2016, observation of progressive erythematous lesions on the lower lips of sun bears from a large captive rescue population in Cambodia led to the detection of SCC and UrHV-1 in a number of individuals. The progressive and potentially debilitating syndrome that follows has welfare and management implications for this large captive population. The potential for an infectious cause of the lesions would result in additional management challenges, as well as important future conservation considerations related to the risk of re-releasing infected individuals to the wild. To better understand the prevalence, risk factors, severity and progression of these lesions, a cross-sectional study was conducted in 2015–2016. We also used histopathology and molecular diagnostics to assess the syndrome in 39 individual sun bears, including the presence of UrHV-1 inclusive of any potential associations with SSC.

## Materials and methods

### Permits

Animal ethics approval was granted by Murdoch University (Permit No. IRMA2850/16). The animal sampling and subsequent analyses in this study were part of the routine

investigation of a clinical condition observed in the animal population. The investigation was conducted by veterinarians responsible for the health and well-being of the animals. No sampling occurred for experimental or purely research purposes; all sampling and testing was indicated with obtain further information about the clinical condition and to drive future efforts to combat its effects on the health of the animals. Permits for international transfer of tissues from a CITES Appendix 1 listed species were obtained for export from Cambodia (No. KH0747, No. KH1059) and import to the United Kingdom (No. 493660/01, No. 543658/01).

## Location and population

This study was conducted at Phnom Tamao Wildlife Rescue Centre (PTWRC), Takeo Province, Cambodia (11˚18'06.5"N, 104˚48'04.7"E). As Cambodia's only government-owned rescue centre PTWRC receives any bears confiscated from the illegal wildlife trade and other exploitative activities. Currently, few options exist for re-release of bears in Cambodia with data lacking on suitable well-protected habitat, and habituation to humans and potential disease risks making it difficult to satisfy accepted guidelines for conservation-purposed release [35]. Consequently, PTWRC currently provides lifelong care to confiscated bears. In 2015, the population comprised 96 sun bears, 38 Asiatic black bears, and one Asiatic black bear x sun bear hybrid. In 2016, the number of sun bears rose to 100. No erythematous lip lesions were observed or previously reported in Asiatic black bears and they were excluded from the study. The hybrid bear [36] showed lesions, and thus was included as a sun bear, bringing the 2015 study population to 97, and to 101 at the 2016 observation period. In 2016, the sun bears ranged from 3 months to 24 years 10 months in age, and comprised 63 females and 38 males.

## Lip lesion prevalence

In both 2015 and 2016 the presence and severity of lip lesions was assessed by visual examination of bears in their dens by one observer, and graded according to a standard scale developed for this study (Fig 1). For cases biopsied before 2015, severity of lip lesions was graded from photographs and clinical notes taken at the time of biopsy, using the same scale. Age and sex were available for each individual. Birth dates are determined by information gathered at the time of each bear's arrival, and are estimated if precise information is not available or known, as is often the case with poached and confiscated adult bears.

## Case series

A sub-set of sun bears (n = 39) had samples collected opportunistically at health examinations between 2011–2016. Because the occurrence of lip lesions was a specific health concern requiring examination and investigation, the majority of sampled individuals had lip lesions. For these individuals, SCC and/or UrHV-1 status were available, based on histological evidence and polymerase chain reaction (PCR) testing.

Thirty-one sun bears had lip lesion biopsies taken for histological examination, with 12 sampled in 2011, and 23 sampled in 2015/16, inclusive of four individuals sampled in both years. PCR to detect UrHV-1 DNA was performed on samples from 31 individual sun bears during 2015/16. This is not the same sub-set for which lip lesion biopsy histology was available, however both PCR and histology results were available for 23 individuals. Eight sun bears had histology only, and another eight had PCR only, bringing the total number of sun bears included in the case series to 39.

| GRADE | LIP LESION GRADE CRITERIA | PHOTO |
|---|---|---|
| 0 | No visible erythematous lesions any-where on the skin of the lower lips | 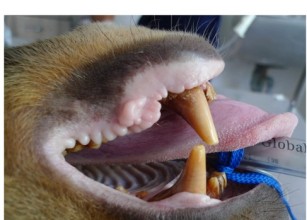 |
| 1 | Focal to multifocal erythematous to erosive-ulcerative lower lip lesions, well to moderately well demarcated, minimally coalescing, and not raised | 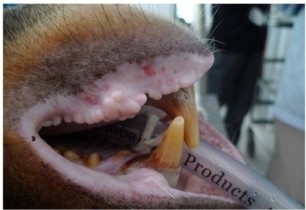 |
| 2 | Multifocal to coalescing or focally extensive erythematous to erosive-ul-cerative lower lip lesions. May be raised or nodular but with <1cm di-ameter. Less than 75% of the total area of the outer lower lip affected | 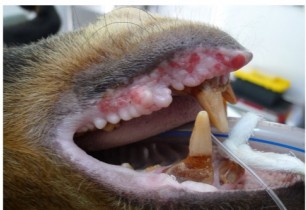 |
| 3 | Multiple coalescing lesions to exten-sive diffuse erythema of the lower lips. Erosive-ulcerative, can be raised or have nodular masses >1cm diame-ter. Greater than 75% of the total area of the lower lips affected | 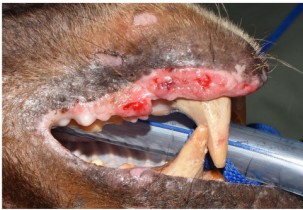 |

**Fig 1. Sun bear (*Helarctos malayanus*) lip lesion severity grades.**

## Sampling methods

Sun bears were immobilised via intra-muscular blow dart (Daninject, Australia) with mede-tomidine (Medetomidine, Troy Laboratories, New Zealand) and zolazepam/tiletamine (Zoletil, Virbac, Australia), administered at a dose rate of 0.0125mg/kg medetomidine and 1.25mg/kg zolazepam/tiletamine. After intubation, anaesthesia was maintained with isoflur-ane (Forane, Baxter Healthcare, USA) and oxygen delivered via a circle re-breathing circuit (CycloFlo, Burton's Medical Equipment, UK). Following the procedure, the medetomidine was reversed with atipamazole (Atipamazole, Troy Laboratories, New Zealand; 0.0625mg/kg), given intramuscularly.

Biopsies were taken from representative areas of lip epithelium, using a 4 to 6mm punch biopsy (Henry Schein, USA). At least one biopsy was placed into each of 10% neutral buffered formalin and viral transport medium (VTM). Biopsy sites were sutured with 3–0 absorbable suture material (Polysorb, Covidien, Ireland). In four cases biopsies were collected from

normal appearing lip skin. In 27 individuals a swab was taken from the oropharyngeal region using a sterile flock-tipped swab (HydraFlock, Puritan Diagnostics LLC, USA) and placed in VTM. Biopsies were opportunistically collected from obvious tongue and vulval lesions, and eight tonsil biopsies were collected. Meloxicam (0.2mg/kg) (Mobic, Boehringer-Ingelheim, USA) was given subcutaneously at the time of biopsy.

## Histopathology

Formalin fixed samples were exported to the International Zoo Veterinary Group laboratory in the United Kingdom. Four biopsies were previously examined at the Animal Health Trust, United Kingdom, in 2012. Tissue sections were routinely processed and embedded in paraffin wax, sectioned at 4 to 6 μm and stained with haematoxylin and eosin. Where necessary, additional special stains were applied, including Periodic Acid Schiff (PAS) reaction to evaluate the basement membrane, Toluidine blue stain for mast cell granules, and Gram stain for bacteria. Tissue sections were evaluated by a board-certified pathologist.

## Virology

Samples in VTM were frozen at -20˚C until transfer to the Institut Pasteur du Cambodge in Phnom Penh. Complete nucleic acids were extracted from biopsy samples using the DNeasy Blood and Tissue kit (Qiagen, Hilden, Germany), according to the manufacturer's instructions. A previously published pan-herpesvirus nested-PCR assay [37], targeting the DNA polymerase (pol) gene, was used to screen the extracted samples for herpesvirus DNA. Round 2 PCR products with a suspected positive amplicon were sent to a commercial provider (Macrogen, Seoul, Republic of Korea) for Sanger capillary sequencing. Contiguous nucleotide sequences were assembled and edited, followed by multiple sequence alignment using Geneious R9 Version 9.1.3. Phylogenetic analyses were conducted on herpesvirus sequences using MEGA X Version 10.05, including selected sequences sourced from the GenBank Database.

## Statistical methods

Lesion observations from 2015 and 2016 were used to calculate the prevalence of lip lesions in the bear population, and a paired Wilcoxon rank sum test was used to test a change in score between the two observation occasions. To investigate the relationship between the dichotomous outcome of lip lesion presence, and sex or age, we used logistic regression. For associations between lip lesion severity grades and sex or age, an ordinal logistic regression was used. The association of these same variables with the score change between 2015 and 2016 was assessed with a mixed effect ordinal logistic regression using R package ordinal [38] with individual bear identification as random effect, and observation occasion, age, and sex as fixed effects. Since the biological sample collection was biased toward bears showing lip lesions, testing for association between UrHV-1 and the presence of lip lesions was not possible. However, on the subset of bears that had both UrHV-1 PCR testing and lesion scores, the association between lesion presence and grade, and UrHV-1 detection was tested using an ordinal logistic regression. The potential association between UrHV-1 detection and score degradation was assessed using the mixed-effect ordinal logistic regression described above.

## Results

### Lip lesion prevalence and risk factors

The overall prevalence of erythematous lip lesions in sun bears was similar between 2015 (64/97, 66.0%) and 2016 (69/101, 68.3%) (Table 1). However, on average, degradation of severity

**Table 1. Observed lip lesion prevalence in sun bears (*Helarctos malayanus*) at a Cambodian rescue centre in 2015 and 2016, including severity grades.**

| Lip lesion grade | 2015 | 2016 |
|---|---|---|
| *Grade 0 (no lesions)* | 33/97 (34%) | 32/101 (31.7%) |
| *Grade 1* | 23/97 (23.7%) | 10/101 (9.9%) |
| *Grade 2* | 27/97 (27.8%) | 33/101 (32.7%) |
| *Grade 3* | 14/97 (14.4%) | 26/101 (25.7%) |
| *Lip lesion any grade* | 64/97 (66%) | 69/101 (68.3%) |

grades was observed between the two observation periods (Wilcoxon paired rank test, V = 48, p<0.0001).

In 2016, the youngest sun bear affected was 3.3 years old and the median age of sun bears affected was 11.9 years. The odds of having lip lesions were increased in older sun bears (OR = 1.5, 95%CI: 1.5–2.6), likewise older bears had increased odds of higher-grade lip lesions (OR = 1.25, 95% CI: 1.17–1.36). Increasing age was also associated with the risk of score degradation between 2015 and 2016 in a mixed effect ordinal logistic regression (OR = 2.04, 95%CI: 1.48–2.81). However sex was not associated with either the presence of lip lesions (OR = 0.7, 95%CI: 0.3–1.6) nor lip lesion severity (OR = 0.92, 95%CI: 0.42–2.03).

## Case series

**Histopathology.** Four of 31 sun bears had repeat biopsies for histopathology, bringing the total number of biopsy cases examined to 35 (Table 2). Every lip section examined had evidence of cheilitis (Fig 2A). A total of 20/31 sun bears (64.5%) had lip SCC confirmed on at least one biopsy, and the age range of sun bears with histological evidence of SCC was 8–20 years. Epithelial changes in the lip tissue ranged from hyperplasia in all cases (n = 35) to epithelial dysplasia (n = 12) (Fig 2B) to frank SCC. Progression of dysplastic changes to SCC in the same biopsy was seen in 7/20 (35.0%) of SCC cases (Fig 2C). Dyskeratotic changes could extend throughout the full thickness of the epidermis. SCC was diagnosed if there was evidence of basement membrane breach and was frequently associated with keratin pearl formation and high mitotic rates (Fig 2C and 2D). Ulceration of the SCCs was a consistent feature, along with inflammation dominated by mononuclear cells with frequent eosinophil and suspected mast cell contribution.

Four cases had lip biopsies taken in 2011 and 2015, and both biopsies were examined as part of this study. Initial biopsies from two of these cases (CR036 and CR038) showed chronic hyperplastic and ulcerative/erosive cheilitis, whereas SCC was seen in the subsequent biopsies. The other two cases (CR026 and CR031) initially had changes consistent with SCC not visible in subsequent biopsies.

Six of nine tongue biopsies submitted for histology showed hyperplastic glossitis, one showed mild multifocal chronic mononuclear perivascular myositis, and two (CR080; CR129) presented as SCC. In one case (CR080), neoplastic epithelial cells were present in all sections of the tongue, and neoplastic islands did not appear to arise from or correspond with the surface epithelium, possibly indicating lingual metastases, although a primary lingual SCC could not be ruled out (Fig 2D).

No infectious pathogen was detected in any of the samples on histological examination with the exception of superficial bacterial colonisation in a small number of animals, which was considered to be secondary to surface erosion.

**Molecular analyses.** Thirty-one sun bears were screened using PCR for UrHV-1 status (Table 2). Twenty-five of those had visible lip lesions; six did not (lip lesion grade 0).

**Table 2. Combined available results of lip histology and Ursid gammaherpesvirus 1 PCR for 39 sun bears (*Helarctos malayanus*) at a Cambodian rescue centre from 2011–2016.**

| Sun bear ID | Age (y) | Lip lesion grade | Lip histology | UrHV-1 any sample | UrHV-1 positive sites | UrHV-1 negative sites |
|---|---|---|---|---|---|---|
| CR023 | 17 | 3 | CH | + | lip lesion (2), vulva | oral swab |
| CR026 | 14 | 3 | SCC (2011) | | | |
| | 18 | 3 | (2015) | - | | lip lesion (2), oral swab |
| CR028 | 16 | 3 | SCC | + | oral swab | lip lesion |
| CR030 | 18 | 2 | SCC | + | lip lesion, tongue lesion | oral swab |
| CR031 | 15 | 3 | SCC (2011)^ | | | |
| | 19 | 3 | CH (2015) | + | lip lesion, tongue lesion | lip lesion, oral swab |
| CR036 | 15 | 3 | CH (2011) | | | |
| | 19 | 3 | SCC (2015) | + | lip lesion, oral swab, tongue lesion, tonsil | lip lesion |
| CR038 | 9 | 2 | CH (2011) | | | |
| | 13 | 2 | SCC (2015) | + | tongue lesion | lip lesion (2), oral swab |
| CR042 | 20 | 2 | SCC (2011)^ | - | | lip lesion, oral swab |
| CR044 | 20 | 2 | CH | + | lip lesion | |
| CR046 | 15 | 2 | SCC | - | | lip lesion, oral swab |
| CR047 | 13 | 2 | CH | - | | lip lesion, normal lip, oral swab, tonsil |
| CR053 | 20 | 1 | CH | - | | lip lesion (2), oral swab |
| CR065 | 11 | 3 | SCC | + | lip lesion (2), tongue lesion | oral swab |
| CR070 | 17 | 2 | SCC | - | | lip lesion (2), oral swab |
| CR077 | 10 | 3 | SCC | - | | lip lesion (2), oral swab |
| CR080 | 10 | 3 | SCC | + | lip lesion | lip lesion, oral swab |
| CR093 | 14 | 2 | SCC | + | lip lesion (2) | oral swab |
| CR104 | 9 | 2 | SCC | + | lip lesion | |
| CR105 | 10 | 3 | SCC | + | tongue lesion | lip lesion (2) |
| CR108 | 13 | 3 | SCC | + | lip lesion | lip lesion, oral swab |
| CR129 | 13 | 3 | SCC | + | tonsil, oral swab | lip lesion, tongue lesion |
| CR139 | 11 | 1 | CH | - | | lip lesion, oral swab |
| CR074 | 11 | 2 | CH | - | | lip lesion, normal lip, oral swab |
| CR006 | 17 | 3 | CH^ | ND | | |
| CR021 | 17 | 3 | CH^ | ND | | |
| CR024 | 14 | 3 | CH | ND | | |
| CR050 | 9 | 3 | CH | ND | | |
| CR058 | 11 | 3 | SCC | ND | | |
| CR095 | 6 | 2 | CH | ND | | |
| CR102 | 8 | 2 | SCC | ND | | |
| CR103 | 10 | 2 | SCC | ND | | |
| CR084 | 11 | 2 | ND | - | - | lip lesion, oral swab, tonsil |
| CR130 | 12 | 0 | ND | + | tonsil | normal lip, oral swab |
| CR131 | 8 | 0 | ND | - | - | normal lip, oral swab, tonsil |
| CR141 | 7 | 1 | ND | - | - | oral swab |
| CR151 | 5 | 0 | ND | + | vulva | |
| CR154 | 6 | 0 | ND | + | tonsil | oral swab |
| CR155 | 5 | 0 | ND | + | tonsil | oral swab |
| CR162 | 4 | 0 | ND | - | - | oral swab |

^histology performed at the Animal Health Trust.

CH = cheilitis; SCC = squamous cell carcinoma; + = PCR positive; − = PCR negative; ND = not done.

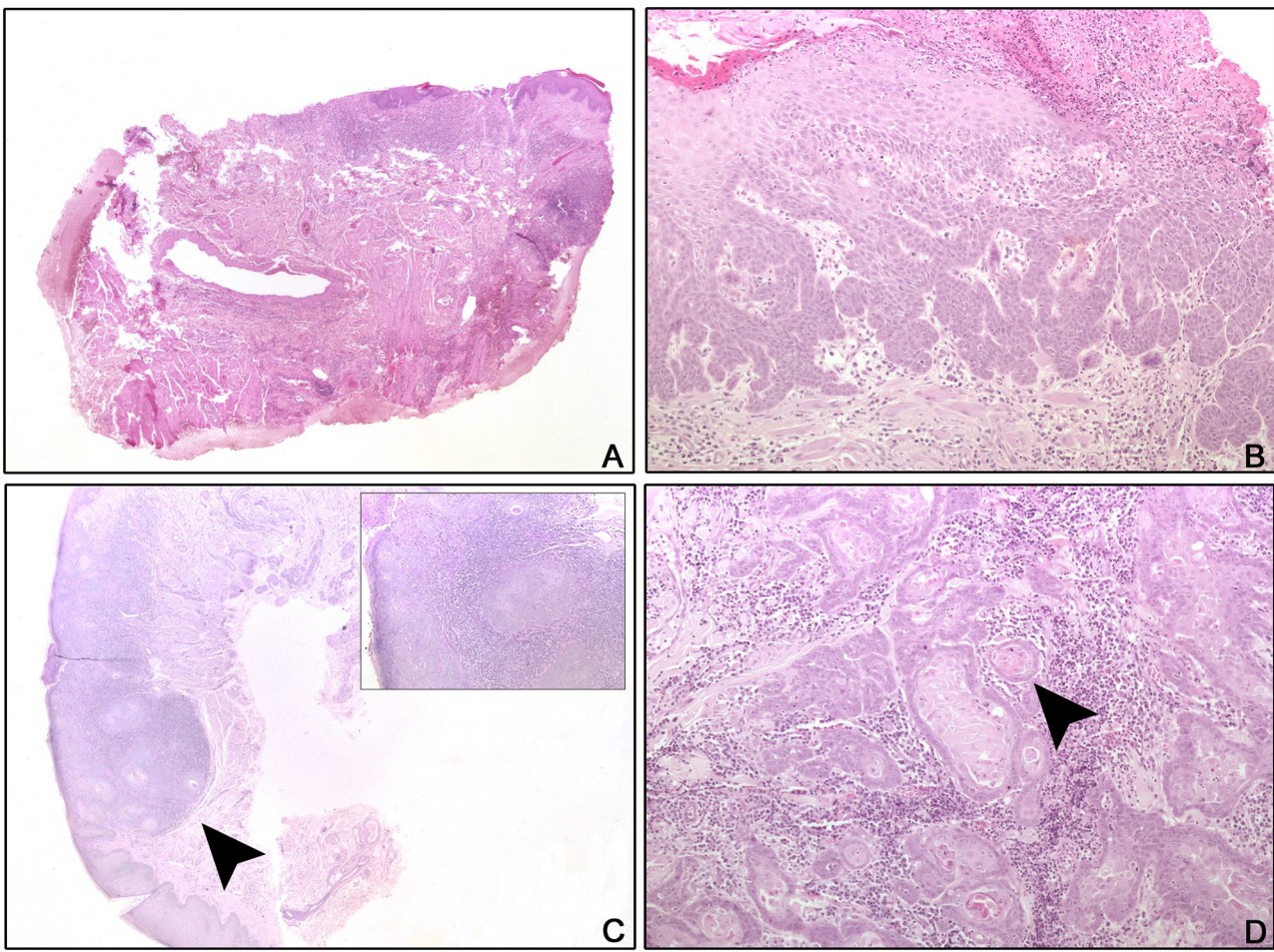

**Fig 2. Histology of lip and tongue lesions from sun bears (*Helarctos malayanus*).** A: CR036 (2011), Lip, H&E, x2: Chronic ulcerative and mildly hyperplastic cheilitis. B: CR026, Lip, H&E, x10: Epithelial hyperplasia and dysplasia with superficial ulceration and inflammation. C: CR036 (2015), Lip, PAS, x2, inset x10: Islands of neoplastic epithelial cells (arrow head) breach the basement membrane (squamous cell carcinoma) and are associated with chronic inflammation. D: CR080, Tongue, H&E, x10: The tongue is widely effaced by cords, trabeculae, and nests of neoplastic squamoid epithelial cells. Keratin pearls are present (arrow head) and the desmoplastic stroma is inflamed.

Combinations of six different sample types were available for testing, including lip lesion biopsy (n = 24), oral swab (n = 27), tongue lesion biopsy (n = 7), tonsil biopsy (n = 8), normal lip biopsy (n = 4), and vulval lesion biopsy (n = 2). Eighteen sun bears (58.1%) had at least one sample positive for UrHV-1.

Of the sun bears with lip lesion tissue biopsied and tested (n = 24), 10/24 (41.7%) were positive for UrHV-1 using PCR on lip lesion tissue. In an ordinal logistic regression adjusting for age, UrHV-1-positive bears had higher lip lesion scores, but the OR confidence interval included 1 (OR = 3.8, 95% CI: 0.98–16.3, p = 0.06). There was no association between the UrHV-1 status and the degradation of lip lesion scores in a mixed effect ordinal logistic regression.

Of the 27 oral swabs collected from sun bears with and without lip lesions, three (11.1%) were positive for UrHV-1 DNA on PCR and were all from sun bears with lip lesions. Four sun bears had biopsies of normal lip tested and two had swabs of lip surface tested; none were positive for UrHV-1 using PCR. Six out of seven tongue lesion samples were positive for UrHV-1. Five out of eight sun bears with tonsil tissue tested by PCR were positive for UrHV-1 DNA,

although only two of the positives (CR036, CR129) had lip lesions. Two sun bears had vulval lesion tissue test positive to UrHV-1. One of these sun bears (CR023) also had lip lesions and tested positive to UrHV-1 on lip lesion biopsy, but her oral swab was negative.

For thirteen sun bears there were two lip lesion biopsies available for duplicate PCR screening. Six of the 13 were negative on both biopsies, three were positive on both, and four were positive on one but negative on the other.

When considering sample type from the 18 sun bears with at least one sample positive for UrHV-1 on PCR, the likelihood of detection varied. Thus, UrHV-1 was detected in 13/22 lip biopsies (59.1%), 3/14 (21.4%) oral swabs, 6/7 (85.7%) tongue lesion biopsies, 5/5 (100%) tonsil biopsies, and 2/2 (100%) vulval biopsies.

Both UrHV-1 and lip SCC status were available for 23 sun bears (Table 2). Of these, 17/23 (73.9%) were diagnosed with lip SCC on at least one biopsy, and of those 12/17 (70.6%) also had a positive UrHV-1 result on PCR including eight (47%) on lip biopsy samples. Two of the 23 individuals (CR023, CR044) were positive for UrHV-1 on lip lesion biopsy but did not have SCC on histology of lip lesion biopsies.

**UrHV-1 phylogenetics.** Sequence analysis of the Round 2 PCR amplicons from the pan-herpesvirus nested-PCR assay revealed that UrHV-1 DNA was amplified from the positive biopsy samples. All of the UrHV-1 pol gene sequences generated from the sun bears (n = 16) were identical (GenBank accession no. MN685118 to MN685133). The highest sequence identity was observed with the UrHV-1 strains detected in sun bears from zoos in the USA (JX220982) [10], from an Asiatic black bear in Russia (MK089801; unpublished) and from free-ranging black bears in the USA [39] (Fig 3). Phylogenetic analysis suggested UrHV-1 is a

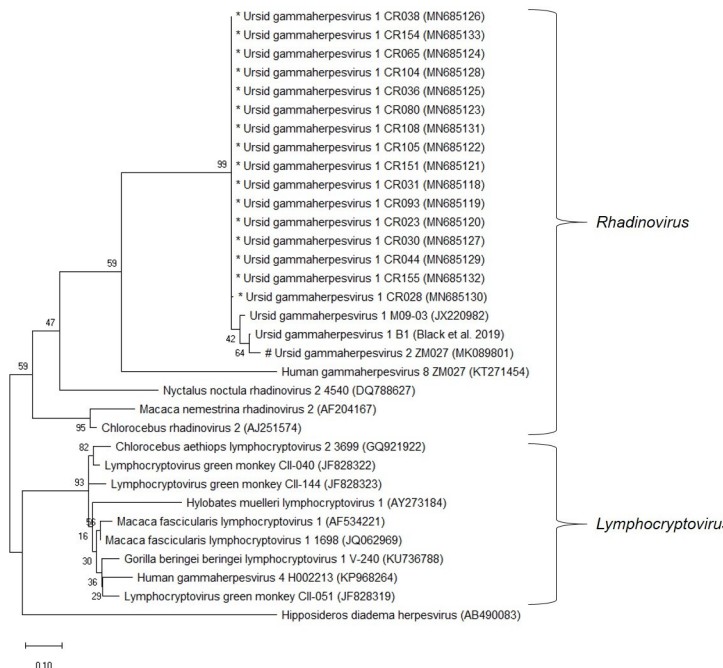

**Fig 3. The evolutionary relationship of Ursid gammaherpesvirus 1 strains compared to representative gammaherpesviruses based on partial DNA polymerase (pol) gene sequences.** The tree was built using the Maximum Likelihood method based on the Tamura 3-parameter model to compute the evolutionary distances. The percentage of replicate trees in which the associated taxa clustered together in the bootstrap test (500 replicates) are shown next to the branches. Evolutionary analyses were conducted in MEGA X (version 10.1.1). * sequences were generated in this study. # As designated by Troyer et al (GeneBank accession: MK089801). However, further sequence data is needed to confirm the designation of a distinct viral species.

member of the subfamily *Gammaherpesvirinae* and in the genus *Rhadinovirus* as previously established.

## Discussion

This study describes the prevalence and histological features of erythematous lip lesions in a captive sun bear population in Cambodia, including data on concurrent infection with a herpesvirus. Phylogenetic analysis, based on the pol gene, supported the classification of this herpesvirus as UrHV-1 in the *Gammherpesviridae* sub family and the *Rhadinovirus* genus, and showed close alignment with other published ursid gammaherpesviruses. Erythematous lip lesions were common in this sun bear population, with a prevalence of 68.3% in 2016, and the presence and severity of lip lesions were significantly associated with increasing age. In addition, older bears were more likely to have a degradation of their lesion scores between the two observation dates. Owing to a biased study population, we were unable to infer a statistical association between UrHV-1 and observed SCCs or lip lesions, however given the biologically plausible links between the virus and neoplastic disease, we recommend this forms a focus for future study design.

Histological examination of lip lesion tissue in the current study revealed cheilitis and hyperplasia in all cases, with progression to SCC in 64.5% of cases. There was frequent histological evidence of progression from dysplasia to SCC in the same section (Fig 2C), suggesting lip lesions are part of a syndrome that progresses to SCC, either inevitably or in the presence of certain co-factors. Chronic inflammatory changes can predispose tissue to neoplasia [5], with SCCs in other species described as developing on a background of inflammation [40–42]. The histological evidence in this study suggests inflammatory change is a significant predisposing factor to lip SCC development. The cause of the underlying lip inflammation is yet to be clarified, however potential causal factors for cheilitis in other species include herpesvirus infection [18,43], UV-induced damage [44], and allergic reaction [45,46]. Future research should aim to incorporate these variables into the study design where possible.

In this study, 70.6% of sun bears with lip SCC had UrHV-1 detected in any sample, including lip lesion biopsy, vulval lesion biopsy, tongue lesion biopsy, tonsil biopsy, and oropharyngeal swab. However, the selection of a subset of sun bears for examination, sampling, and UrHV-1 testing was strongly biased towards those showing lip lesions. As a result, we were not able to reliably assess the association between UrHV-1 infection and lip lesion or SCC development. Similarly, Lam et al [10] found UrHV-1 in 4/5 (80%) of cases of oral SCC, but did not present results on healthy individuals. Additional opportunistic sampling of apparently healthy populations should be pursued to improve our understanding of the association between UrHV-1 and sun bear lip lesions or SCC, and whether the virus is ubiquitous in sun bears. Importantly, while we focussed on detection of UrHV-1 in lip lesion tissue, our results show that the virus is detectable on oral swabs and in vulval and tonsillar tissue from sun bears without lip lesions. This highlights the potential for non-invasive sampling of healthy populations, particularly if the sensitivity of virus detection testing can be increased, and ideally paired with serological test development.

A defining property of gammaherpesviruses is the ability to cause cancer, facilitated by certain co-factors [29,47,48], confirming the biological plausibility of UrHV-1 induced SCC in sun bears. Yet the identification of UrHV-1 from lip SCC tissue was inconsistent, with only 47% of lip SCC biopsies testing positive for UrHV-1. Consideration should be given to a "hit-and-run" mechanism, where viral DNA responsible for initiating transformation of cells may no longer be detectable in tumour tissues, as has been hypothesised for the role played by gammaherpesviruses in some human tumours [28]. Likewise, the sensitivity of PCR testing and/or

the biopsy site selection might have affected results. This is evidenced by the discordant UrHV-1 PCR results for four sun bears that had duplicate lip lesion biopsies taken on the same day and tested separately, and the relatively low proportion (59.1%) of UrHV-1 positive lip lesion biopsies from sun bears that were positive on any sample. Development of a real-time PCR assay to improve sensitivity may increase virus detection rates. Serial screening of individuals utilising multiple samples from a range of sites could also improve sensitivity of detection and guide the development of an effective sampling strategy to reliably detect UrHV-1 in infected sun bears. Similarly, it was surprising that two cases had lip SCC when punch biopsied in 2011 but biopsies in 2015 did not show neoplasia, possibly also reflecting the insensitivity of the biopsy sampling and virus detection methods used.

Describing the full genome of UrHV-1 is a priority to better define the role, if any, played by the virus in sun bear lip SCC. This will increase our understanding of the phylogeny, and possibly identify differences in genus-specific regions in the genome indicating the existence of multiple ursine gammaherpesviruses, with potential differences in pathogenicity [34]. Analysis of additional genome is necessary to enable in-situ hybridisation, the gold standard method for confirming the presence of virus in tumour cells [49]. Further, the full genome may clarify the oncogenic potential of the virus, for example by looking for the unique open reading frames found in other oncogenic gammaherpesviruses at the 5' end of the genome [29,50].

Latency is a hallmark property of herpesviruses, with complex modification of the host immune response allowing virus to persist virtually undetected in infected cells [51,52]. The specific cellular site of latency for UrHV-1 in sun bears, if this occurs, is not known. Virus detection in over half the tonsil samples tested (5/8; 62.5%) suggests this could be an important site for virus replication or latency, particularly given gammaherpesviruses are known to be highly lymphotropic [29]. Although potentially a result of salivary contamination, the detection of UrHV-1 in vulval lesions was interesting given gammaherpesviruses have been found in association with genital lesions in other species including Eastern grey kangaroos (*Macropus giganteus*)[53], bottle-nosed dolphins (*Tursiops truncates*) [54], and California sea lions (*Zalophus californianus*) [55]. Ursid gammmaherpesvirus 1 was detected in 6/7 tongue lesions biopsied, although no sampling of normal tongues was carried out, and salivary contamination is possible. Further work, such as using immunohistochemistry to detect specific viral proteins, and using additional viral genome to design a real time PCR assay to increase test sensitivity, will help determine whether UrHV-1 is detectable in various tissues.

A major risk factor for the progression of actinic cheilitis to lip SCC in people is chronic dose-dependent exposure to solar radiation [56–58]. The outer lip, or vermillion in human anatomy, has a thinner epithelium, less keratin, and lower melanin content compared to skin, rendering it vulnerable to sun damage [57,59]. Solar radiation has also been implicated in reactivated herpes simplex virus 1 lip lesions in people due to local UV radiation induced immunosuppression [60,61]. The lower lips of the sun bear hang below the cover of the upper lips, and panting to dissipate heat further exposes the pale hairless skin to sunlight. The nature of outdoor enclosures means captive sun bears are likely exposed to more solar radiation than in the canopied forests of their natural habitat. The role of UV exposure in sun bear lip lesions could be explored through a prospective cohort study, with enclosure adaptations to reduce UV exposure in a group of bears with similar risk profiles. Additionally, the use of histological stains to indicate UV damage such as solar elastosis [62] will help infer the role played by sun exposure.

The presence of eosinophils and mast cell populations in histological examination of the sun bear lip lesions highlights a possible allergic aetiology. Allergy is a frequently reported cause of non-actinic cheilitis in humans [63]. While this warrants consideration, the individual determinants of allergic disease susceptibility [64] make it unlikely as a cause of such a

prevalent syndrome in this population. Furthermore, there are indications in the human literature that eosinophils and mast cells in fact play a role in the development of tumour angiogenesis in the progression of actinic cheilitis to lip SCC [65,66] which gives weight to a UV exposure rather than allergic aetiology.

Until data is available on the prevalence of UrHV-1 in wild sun bears, it remains unknown if captive conditions contribute to the spread of the virus and/or its role in lip lesions and their progression. As well enclosures being relatively sun exposed, rescue centres often have unavoidably high stocking rates, facilitating prolonged close contact and disease transmission. Additionally, wildlife species experience stress in captive environments and can develop persistent adrenocortical responses that may exert physiological effects on the immune system [67]. Reactivation of herpesviruses in immune–compromised individuals contributes to oral lesions in other species [19,52]. Compromised host immunity could also contribute to the transformation of lesions into SCC, with human transplant recipients on long-term immuno-suppressive therapy having an up to 9-fold increase in premalignant and malignant lip lesions [68]. The relationship between host-immunity, UrHV-1 detection, and the progression and transformation of lip lesions warrants further exploration.

## Conclusions

Together these preliminary descriptive data provide a strong basis to drive further research into the potential cause(s) of sun bear lip lesions, their progression to SCC, and the role, if any, played by UrHV-1 with or without other co-factors. Defining this syndrome further will allow better management of affected captive populations, through the development of prevention and treatment initiatives. Furthermore, there are important conservation implications for this threatened species, with the progressive nature of the syndrome likely to affect long-term health and therefore suitability of individuals for re-release programmes. The study highlights the opportunity afforded by large wildlife rescue collections to promote a better understanding of the veterinary health consequences of captivity, as well as to identify and address disease susceptibility in relatively understudied wildlife species in order to better assess potential health risks to wild populations.

## Supporting information

**S1 Dataset.**
(XLSX)

## Acknowledgments

We thank the Forestry Administration of Cambodia and Free the Bears for their support of the veterinary programme at Phnom Tamao Wildlife Rescue Centre where this work was conducted. We thank the Animal Health Trust for performing the initial histological examination of four sun bear lip lesion biopsies in 2012.

## Author Contributions

**Conceptualization:** Kirsty Officer, Nhim Thy, Nev Broadis, Bethany Jackson.

**Data curation:** Kirsty Officer, Vibol Hul.

**Formal analysis:** Kirsty Officer, Mathieu Pruvot, Paul Horwood, Bethany Jackson.

**Funding acquisition:** Mathieu Pruvot, Paul Horwood, Philippe Dussart.

**Investigation:** Kirsty Officer, Paul Horwood, Daniela Denk, Vibol Hul.

**Methodology:** Kirsty Officer, Mathieu Pruvot, Paul Horwood, Daniela Denk, Kris Warren, Bethany Jackson.

**Project administration:** Kirsty Officer, Nev Broadis.

**Resources:** Nhim Thy, Nev Broadis.

**Supervision:** Paul Horwood, Kris Warren, Nhim Thy, Philippe Dussart, Bethany Jackson.

**Visualization:** Kirsty Officer, Paul Horwood, Daniela Denk.

**Writing – original draft:** Kirsty Officer.

**Writing – review & editing:** Mathieu Pruvot, Paul Horwood, Daniela Denk, Kris Warren, Vibol Hul, Nev Broadis, Philippe Dussart, Bethany Jackson.

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
