## [Decision Letter · Decision Letter 0]

6 Oct 2020

PONE-D-20-19581

Epidemiology and pathological progression of erythematous lip lesions in captive sun bears (Helarctos malayanus)

PLOS ONE

Dear Dr. Officer,

Thank you for submitting your manuscript to PLOS ONE. After careful consideration, we are pleased to inform you that your manuscript has been judged scientifically suitable for publication but does not fully meet PLOS ONE’s publication criteria as it currently stands. Therefore, we invite you to submit a revised version of the manuscript together with a rebuttal letter that addresses the comments as stated below.

We look forward to receiving your revised manuscript.

Kind regards,

Kristin Mühldorfer

Academic Editor

PLOS ONE

Journal Requirements:

Reviewers' comments:

Reviewer's Responses to Questions

**Comments to the Author**

1. Is the manuscript technically sound, and do the data support the conclusions?

Reviewer #1: Yes

Reviewer #2: Yes

2. Has the statistical analysis been performed appropriately and rigorously? 

Reviewer #1: Yes

Reviewer #2: Yes

3. Have the authors made all data underlying the findings in their manuscript fully available?

Reviewer #1: Yes

Reviewer #2: Yes

4. Is the manuscript presented in an intelligible fashion and written in standard English?

Reviewer #1: Yes

Reviewer #2: Yes

5. Review Comments to the Author

Reviewer #1: The authors report on their investigations on erythematous to neoplastic lip lesions in 39 sun bears kept in captivity in a rescue center in Cambodia with a total number of 100 bears.

They documented the progression of the lesions and compared histology results from biopsies of these pathological changes with molecular investigations on gammaherpesvirus Ursid herpesvirus 1 of the same lesion as well as additional virus sampling sites. Previously Lam et al. described a similar study on a small number of captive sun bears from North American zoological institutions, where UrHV-1 was discovered.

The manuscript is well written and structured and the detailed description of the results highlights the puzzling outcome of the study, clearly urging further investigations. The authors discuss their findings with enough caution not to over interpret their sometimes contradicting findings.

Clearly the content of the study is of importance for the scientific community working in the field of zoo medicine and animal conservation and ideally should prompt investigations in a larger set of animals.

Just out of personal interest and acknowledging that Giemsa staining was also performed on histologic sections – did the authors consider Treponema sp. as a possible cause, which sometimes do not easily show in Giemsa ? Somehow the sun bear lesions reminded me of Treponema cuniculi lesions in rabbits…

Reviewer #2: Sunbeam paper plosone-

This is an informative, well-written, and sound descriptive study of herpesvirus-associated lip lesions potential squamous cell carcinoma in sun bears. This has been an emerging issue in these animals in captivity, so it warrants attention.

Discussion-

Line 381 regarding inconsistent results with pcr- (this is a comment not a critique- you might want to try laser-dissection capture microscopy of selected foci or cell types or areas of lesions most likely to have virus. This can often help with detection)-

Immunohistochemistry may be useful in these cases as well (see the work done on otarid herpesvirus I-it cross reacted with EBV monoclonal antibodies on IHC)

EM also can improve detection. So, I also recommend you mention IHC and EM as ancillary diagnostics that may be helpful.

Additionally, you may want to develop a qPCR method which can be more sensitive (you mention sensitivity here already). Further,

In terms of the comment “Similarly, it was surprising that two cases had lip SCC in 2011 but biopsies in 2015 did not show neoplasia”, was surgical excision done of the site ? If so, it might have impacted the diagnosis.

Has there been any evidence of metastases in any of these cases?

line 404-4-5 please also reference work and mention otarine herpesvirus associated with genital lesions/tumors in California sea lions and add.

6. PLOS authors have the option to publish the peer review history of their article (what does this mean?). If published, this will include your full peer review and any attached files.

Reviewer #1: No

Reviewer #2: No

---

## [Author Response · Author response to Decision Letter 0]

12 Nov 2020

All reviewer comments are addressed in the "Response to Reviewers" letter attached

---

## [Editor Report · Decision Letter 1]

17 Nov 2020

Epidemiology and pathological progression of erythematous lip lesions in captive sun bears (Helarctos malayanus)

PONE-D-20-19581R1

Dear Dr. Officer,

We’re pleased to inform you that your manuscript has been judged scientifically suitable for publication and will be formally accepted for publication once it meets all outstanding technical requirements.

Kind regards,

Kristin Mühldorfer

Academic Editor

PLOS ONE

---

## [Editor Report · Acceptance letter]

20 Nov 2020

PONE-D-20-19581R1 

Epidemiology and pathological progression of erythematous lip lesions in captive sun bears (*Helarctos malayanus*) 

Dear Dr. Officer:

I'm pleased to inform you that your manuscript has been deemed suitable for publication in PLOS ONE. Congratulations! Your manuscript is now with our production department. 

Kind regards, 

on behalf of

Dr. Kristin Mühldorfer 

Academic Editor

PLOS ONE